# Surface Modification of a Graphite Felt Cathode with Amide-Coupling Enhances the Electron Uptake of *Rhodobacter sphaeroides*

**Hana Nur Fitriana** [1,†], **Jiye Lee** [1,†], **Sangmin Lee** [1], **Myounghoon Moon** [1], **Yu Rim Lee** [1,2], **You-Kwan Oh** [3], **Myeonghwa Park** [3], **Jin-Suk Lee** [1], **Jinju Song** [1] and **Soo Youn Lee** [1,*]

1   Gwangju Bio/Energy R&D Center, Korea Institute of Energy Research, Gwangju 61003, Korea; hana3004ap@gmail.com (H.N.F.); jiyelee@kier.re.kr (J.L.); silverlsm@kier.re.kr (S.L.); mmoon@kier.re.kr (M.M.); rimin9@kier.re.kr (Y.R.L.); bmjslee@kier.re.kr (J.-S.L.); jinju@kier.re.kr (J.S.)
2   Interdisciplinary Program of Agriculture and Life Sciences, Chonnam National University, Gwangju 61186, Korea
3   School of Chemical Engineering, Institute for Environmental & Energy, Pusan National University, Busan 46241, Korea; youkwan@pusan.ac.kr (Y.-K.O.); myeonghwa@pusan.ac.kr (M.P.)
*   Correspondence: syl@kier.re.kr; Tel.: +82-62-717-2436
†   These authors contributed equally to this work.

**Abstract:** Microbial electrosynthesis (MES) is a promising technology platform for the production of chemicals and fuels from $CO_2$ and external conducting materials (i.e., electrodes). In this system, electroactive microorganisms, called electrotrophs, serve as biocatalysts for cathodic reaction. While several $CO_2$-fixing microorganisms can reduce $CO_2$ to a variety of organic compounds by utilizing electricity as reducing energy, direct extracellular electron uptake is indispensable to achieve highly energy-efficient reaction. In the work reported here, *Rhodobacter sphaeroides*, a $CO_2$-fixing chemoautotroph and a potential electroactive bacterium, was adopted to perform a cathodic $CO_2$ reduction reaction via MES. To promote direct electron uptake, the graphite felt cathode was modified with a combination of chitosan and carbodiimide compound. Robust biofilm formation promoted by amide functionality between *R. sphaeroides* and a graphite felt cathode showed significantly higher faradaic efficiency (98.0%) for coulomb to biomass and succinic acid production than those of the bare (34%) and chitosan-modified graphite cathode (77.8%), respectively. The results suggest that cathode modification using a chitosan/carbodiimide composite may facilitate electron utilization by improving direct contact between an electrode and *R. sphaeroides*.

**Keywords:** microbial electrosynthesis; cathode; amide-coupling; *Rhodobacter sphaeroides*; $CO_2$

## 1. Introduction

Microbial electrosynthesis (MES) is an economically emerging bio-electrochemical technology for transforming $CO_2$ and renewable electrical energy into chemicals and fuels [1,2]. It could allow the storage and increase in value of intermittent renewable energies such as solar and wind [3]. In an MES reaction, the electroactive microorganisms (called electrotrophs) can utilize electrons ($e^-$) from external conducting materials (i.e., electrodes) as reducing energy to catalyze the conversion of $CO_2$ [4]. These electrotrophs achieve $CO_2$ conversion through several carbon-fixation pathways including the reductive pentose phosphate cycle (e.g., photo- or chemoautotrophs), reductive tricarboxylic acid cycle (e.g., *Clostridium thiosulfatophilum*), and reductive acetyl-CoA pathway (e.g., *Clostridium ljungdahlii*) [1,5]. *Rhodobacter sphaeroides*, a $CO_2$-fixing chemoautotrophic bacterium, can catalyze both the production and consumption of hydrogen molecules ($H_2$) [6]. In a recent investigation, MES-driven $CO_2$ uptake and $H_2$ production in *R. sphaeroides* were found to occur simultaneously without additional organic carbon substrates [7].

Extracellular electron uptake mechanisms of electrotrophs from electrodes rely on: (i) indirect electron uptake by $H_2$ or electron shuttles (i.e., natural and artificial redox

mediators) or (ii) direct electron uptake via physical contact through electron-transfer proteins or apparatus [1,8]. Although the cathodically evolved $H_2$ can support electrotrophic growth of a microorganism, the direct supply of cathodic electrons to microorganisms is more sustainable and energy efficient in an MES reaction [8]. Direct electron uptake from electrodes occurs in range of more positive potential ranges ($-0.4$ to $-0.5$ V vs. SHE) than $H_2$-mediated indirect electron uptake [4,8]. Meanwhile, using artificial redox mediators has drawbacks to application due to their chemical instability, toxicity to microbes, and difficulty in separating them from electrolytes [9]. A dense and well-developed cathodic biofilm is a key parameter for improving direct electron uptake rates and production rates in MES [10]. For example, an increased biofilm of *Sporomusa ovata* on a graphene-functionalized carbon composite cathode enabled higher cathodic current consumption with higher acetate production compared with a bare cathode in an MES reactor [11]. Furthermore, accelerating robust biofilm formation enabled improvement in MES performance. Zhang et al. reported that a modified cathode with positively charged layers such as chitosan, cyanuric chloride, 3-aminopropyltriethoxysilane, and polyaniline showed enhanced electron consumption and acetate production rates of *S. ovata* in an MES reactor [12]. This is because a cathode surface coated with positively charged layers is more suitable for interactions with negatively charged microorganisms [13].

In this study, we performed the cathode modification for MES by *R. sphaeroides* to increase bioproducts synthesis and Faradaic efficiency. In order to promote direct electron uptake, the graphite felt material was coated with a chitosan layer. The chitosan layer presents unique functionalities (co-existing polycationic and neucleophilic properties) that are suitable for constructing a rigid biofilm onto the electrode surfaces [13]. Additionally, the chitosan layer was modified with a carbodiimide compound to enable more substantial interaction between the *R. sphaeroides* cells and cathode surface (Figure 1).

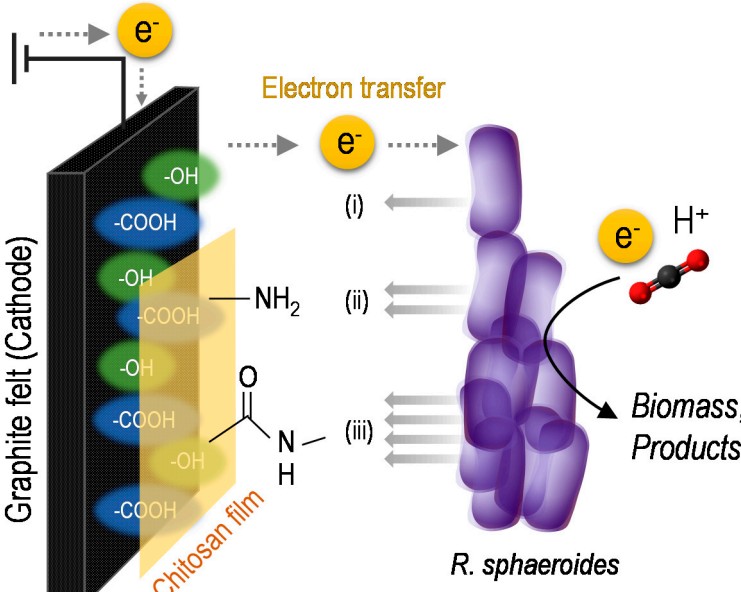

**Figure 1.** Schematic illustration of the proposed mechanisms of bacterial adsorption (*Rhodobacter sphaerodies*) onto the graphite felt electrode (GFE). (i) Bacterial adsorption onto the bare (GFE), (ii) chitosan-modified (GFE/Chit), and (iii) chitosan/carbodiimide compound-modified (GFE/Chit/CDI) cathode. Electron and proton ($H^+$) sources for microbial $CO_2$ conversion of *R. sphaeroides* can be supplied from the cathode part via the anodic water splitting reaction.

## 2. Materials and Methods

### 2.1. Strain and Medium Preparation

The *Rhodobacter sphaeroides* 2.4.1 strain was obtained from the Korea Collection of Type Cultures (KCTC, strain no. 1434). The activation and growth medium had the following

composition (L$^{-1}$): 2.72 g KH$_2$PO$_4$, 1.95 g NH$_4$Cl, 4 g of succinic acid, 0.1 g of glutamic acid, 40 mg of aspartic acid, 0.5g NaCl, 0.2 g of nitriloacetic acid, 0.3 g MgSO$_4$·7H$_2$O, 33.4 mg CaCl$_2$·2H$_2$O, 2 mg FeSO$_4$·7H$_2$O, 2 mg (NH$_4$)$_6$Mo$_7$O$_{24}$, 100 μL of a trace elements solution, and 100 μL of a vitamins solution. Pure carbon dioxide was used as the sole carbon source for the microbial electrosynthesis reaction instead of succinic acid. The trace elements solution had the following composition (100 mL$^{-1}$): 1.765 g EDTA, 10.95 g ZnSO$_4$·7H$_2$O, 5.0 g FeSO$_4$·7H$_2$O, 1.54 g FeSO$_4$·7H$_2$O, 1.54 g MnSO$_4$·H$_2$O, 0.392 g CuSO$_4$·5H$_2$O, 0.284 g Co(NO$_3$)·6H$_2$O, and 0.11 g H$_3$BO$_3$. The vitamins solution had the following composition (100 mL$^{-1}$): 1 g of nicotinic acid, 0.5 g of thiamin HCl, and 0.1 g of biotin. The medium was adjusted to pH 7.0 using a 20% KOH solution. All culture media were sterilized by filtration through a hydrophilic membrane filter with a pore size of 0.2 μm (Adventec Ltd., Tokyo, Japan).

### 2.2. Microbial Electrosynthesis Reactor Configuration and Operation

A double-chamber H-type reactor was used for the microbial electrosynthesis (MES) reaction (working volume: 350 mL) (Figure 2). The anode and cathode chambers were joined by a glass arm (38 mm diameter) and separated using a proton exchange membrane (PEM, Nafion 117; DuPont Ltd., Wilmington, DE, USA). The anode and cathode electrodes were pieces of graphite felt, and were 4 cm × 10 cm and 4 cm × 5 cm, respectively. The thickness of each electrode was 0.3 cm (GF030, FuelCellStore, College Station, TX, USA), and each one was connected to titanium wire which acted as the current collector from anode to cathode. The reference electrode, Ag/AgCl (in 3.0 M NaCl, 0.209 V vs. NHE) was placed in the cathode chamber. All bottles and reactors were sterilized at 121 °C for 15 min by autoclaving. After autoclaving, the cathode chamber was flushed with filtered 5% CO$_2$ (balanced with Ar) for at least 1 h to remove any oxygen remaining in the graphite felt.

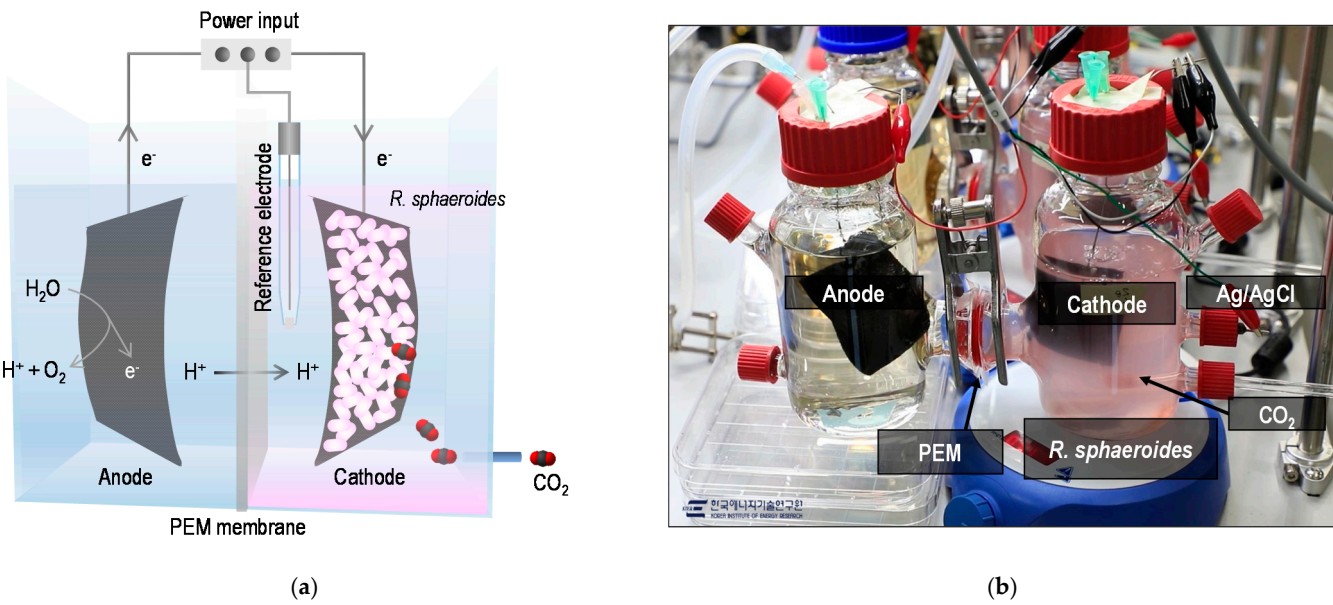

(**a**)          (**b**)

**Figure 2.** (**a**) Schematic representation and (**b**) Photograph of the microbial electrosynthesis reactor used for conversion of CO$_2$. CO$_2$ conversion was performed in growth medium (pH 7.0) without a carbon source other than supplementation with 5% CO$_2$ at an applied potential of −0.6 V (vs. Ag/AgCl).

The cathode electrode was poised with −0.6 V (vs. Ag/AgCl) using a potentiostat (WMPG1000; WonAtech, Seoul, Korea). The experiments were carried out under anaerobic conditions in batch mode. The cathode chamber was supplied with 5% CO$_2$ (*v/v*) balanced with Ar at a rate of 17.5 mL min$^{-1}$ (0.05 vvm). All MES reactors were placed under white LED lamps in the range of 450–475 nm (www.photonic.com) at 28 ± 3 °C, which provided 50 μmol photons m$^{-2}$ s$^{-1}$. The reactors were operated in triplicate.

### 2.3. Electrode Modification Procedure

The graphite felt was pretreated with 5% HCl (*v/v*) overnight, then thoroughly washed with Milli-Q water (18.2 MΩ cm) and dried at 60 °C overnight. For cyclic voltammetry analysis, a glassy carbon electrode was polished with aluminum–water slurries (diameter of 0.3 μm and 1 μm). To fabricate the chitosan-modified electrode, graphite felt of glassy carbon was immersed in 2% chitosan (50–190 kDa, *v/v*) solution dissolved in 2% acetic acid (*v/v*) overnight, then dried overnight at 60 °C. To fabricate the chitosan/carbodiimide (CDI) compound-modified electrode, the dried electrode was immersed in ethanol–water (4:1, *v/v*) coupling medium containing 1-ethyl-3-(3-dimethylaminopropyl) carbodiimide and *N*-hydroxysuccinimide (50 mM/50 mM) overnight at room temperature, then carefully washed with ethanol and dried overnight at room temperature.

### 2.4. Analytical Methods

To measure the total biomass of planktonic cells, optical density (OD) was measured using a UV/Vis spectrophotometer (Effendorf Biospectrometer, AG., Hamburg, Germany), and then corrected to biomass for *R. sphaeroides* to determine the amount of the dry weight. A biomass-to-$OD_{660}$ correlation for *R. sphaeroides* of 0.55 g $L^{-1}$ cell dry weight per unit $OD_{600}$ was obtained (data not shown). The quantity of attached bacterial calls onto the graphite felt cathode were determined by a gravimetric method for each biocathode. The weight of dried cathode, before and after MES operation, was measured. A mass change of dried cathode after MES operation was calculated as biofilm weight.

The zeta-potential of *R. sphaeroides* and the graphite felt electrodes was measured using a Zetasizer (Nano-ZS92; Malvern Panalytical Ltd., Malvern, UK). All samples were analyzed in 0.1 M sodium phosphate buffer (pH 7.0). All potentials were measured against an Ag/AgCl reference electrode (3.0 M NaCl, Bioanalytical systems, West Lafayette, IN, USA). Cyclic voltammetry was performed using a potentiostat (SP-200; Biologics, Paris, France). Oxygen was removed from the electrolyte by bubbling with oxygen-free $N_2$ for 10 min before electrochemical measurements. The applied scanning rate was 20 mV $s^{-1}$ within the ranges of −0.3 to 0.6 V and −0.4 to 0.2 for $[Fe(CN)_6]^{3-}$. Sodium phosphate buffer (0.1 M at pH 7.0) was used as the electrolyte. The microscopic features of the samples were investigated using a MIRA 3 LMU model scanning electron microscope (TESCAN, Czech Rep.) at an accelerating voltage of 10 kV. For liquid chromatography analysis, medium samples and standards were filtered through a syringe filter with pore size of 0.2 μm (Adventec Ltd., Tokyo, Japan). The filtrate of the samples was analyzed by a high-performance liquid chromatography (Agilent 1260; Agilent Technologies, Santa Clara, CA, USA) equipped with an HPX-87 H column.

### 2.5. Calculation of Faradaic Efficiency

The efficiency values reported in this work are based on the statistical average of at least three biological replicates. When the biocathodes operated at −0.6 V (vs. Ag/AgCl, saturated NaCl) showed steady or repeatable current, the biocathodes were operated at fixed constant potential to measure the biomass and succinic acid production, and the faradaic efficiency. To measure the total biomass of the planktonic cells, optical density (OD) was corrected to biomass for *R. sphaeroides* to determine the dry weight. Cells attached to the biocathodes were determined by a gravimetric method for each biocathode. The production of succinic acid was measured by a high-performance liquid chromatography (Agilent 1260, Agilent Technologies, Santa Clara, CA, USA) equipped with an HPX-87 H column. The faradaic efficiency (η) was calculated according to the following Equation (1) [14]:

$$\eta = (mnF)/(\int_{t=0}^{t} Idt) \times 100 \tag{1}$$

where *m* denotes the mole of products (mol), *n* denotes the number of electrons transferred for the production of 1 mole of products (succinic acid), *F* denotes the faradaic constant (96,485 C $mol^{-1}$), *I* denotes the current (A), and *t* denotes the time (s).

Biomass formation:

$$CO_2 + 0.095(NH_4)_2SO_4 + 4.42H^+ + 4.42e^- \rightarrow$$
$$CH_{1.99}O_{0.50}N_{0.19} + 0.095H_2SO_4 + 1.5H_2O \tag{2}$$

Succinic acid formation:

$$4CO_2 + 6H_2O + 3.5e^- \rightarrow C_4H_6O_4 + 3.5O_2 + 3H_2O \tag{3}$$

## 3. Results and Discussion

### 3.1. Modification of the Graphite Felt Electrode for Amide-Coupling

The graphite materials were used as the substrate to sustain direct supplementation of electrons to the bacteria [15,16]. Electron uptake from extracellular sources is mainly dependent on direct cell contact via a biofilm on the cathode surface [17]. Modification of cathode materials by decorating with functional groups and increasing surface hydrophilicity might benefit the adhesion and growth of bacterial biofilm [1]. Generally, negatively charged functional groups on the bacterial surface are responsible for the electrostatic binding of cationic groups [18]. However, the surface charge of pristine carbon-related materials is neutral [12]. To generate a positively charged electrode surface, aminopolysaccharide chitosan polymer was used to coat the graphite felt surface by immersion. In addition, to induce covalent attachment of bacteria to the electrode surface, the chitosan-coated graphite felt was further modified with carbodiimide compound to provide robust amide functionality [19]. The possible chemical pathway of amide-coupling between chitosan and bacteria through the carbodiimide cross-linker is described in Figure 3 [20].

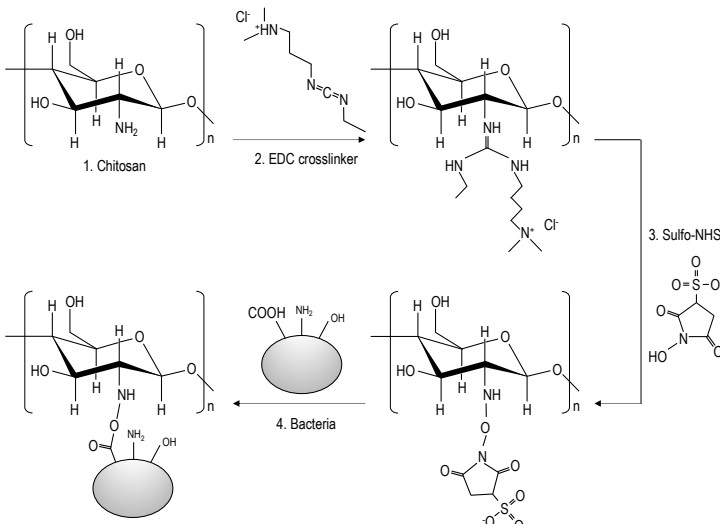

**Figure 3.** Reaction of the carbodiimide compound (EDC) with chitosan free amines (1,2). The intermediate is stabilized by the formation of sulfo-NHS (3) which undergoes nucleophilic substitution to form an amide bond with the carboxyl groups on the bacteria (4).

To estimate bacterial adhesion to the modified electrodes, the zeta-potential of *R. sphaeroides* and powdered graphite felt electrodes were measured in a neutral environment (0.1 M sodium phosphate, pH 7.0) (Table 1). The zeta-potentials of both *R. sphaeroides* ($-31.88 \pm 3.31$ mV) and bare graphite felt ($-10.13 \pm 2.72$ mV) were negative. Meanwhile, the zeta-potential of the graphite felt surface was shifted to positive range by the modification with chitosan and the chitosan/carbodiimide compound. This suggests that the modified graphite felt may form a more rigid biofilm of *R. sphaeroides* than bare graphite felt due to improved bacterial adhesion [21].

**Table 1.** Zeta-potential results for *R. sphaeroides*, bare graphite felt, chitosan-modified graphite felt, and chitosan/carbodiimide compound-modified graphite felt in 0.1 M sodium phosphate buffer (pH 7.0).

|  | $\zeta$-**Potential (mV)** |
|---|---|
| *Rhodobacter sphaeroides* | $-31.88 \pm 3.31$ |
| Bare graphite felt | $-10.13 \pm 2.72$ |
| Graphite felt/chitosan | $1.81 \pm 0.61$ |
| Graphite felt/chitosan/carbodiimide | $1.74 \pm 0.72$ |

### 3.2. Electrochemical Evaluation of Chitosan- and Chitosan/Carbodiimide Compound-Modification

The electrode consisted of chitosan and carbodiimide compounds immobilized on the graphite felt surface. Prior to combining the modified electrode with the MES reactor, we checked the redox reactions of the chitosan- and carbodiimide compound-modified layers by cyclic voltammetry with electroactive species. Figure 4a shows the cyclic voltammograms (CVs) of a bare and a modified glassy carbon electrode (GCE) in 1 mM $[Fe(CN)_6]^{3-}$ solution. In this experiment, all the electrodes were examined in 0.1 M sodium phosphate buffer (pH 7.0). The separation between the anodic ($E_{pa}$) and cathodic ($E_{pc}$) peak potentials ($\Delta E_p$) of bare GCE in $[Fe(CN)_6]^{3-}$ solution was 0.28 V. On the other hand, the $\Delta E_p$ values of chitosan and of chitosan/carbodiimide compound-modified GCE decreased to 0.23 and 0.11 V, respectively. The peak current densities ($J_p$) of the modified electrodes in the CVs increased and the midpoint potentials ($E_m$) shifted to the negative with the modification using chitosan and the chitosan/carbodiimide compound. $[Fe(CN)_6]^{3-}$ ions might have been confined in positively charged layers (Table 1) through the force of electrostatic attraction [22]. From these results, we presume that the electrode coated with amine and amide functional groups might have facilitated the migration of negatively charged *R. sphaeroides* cells toward the electrode.

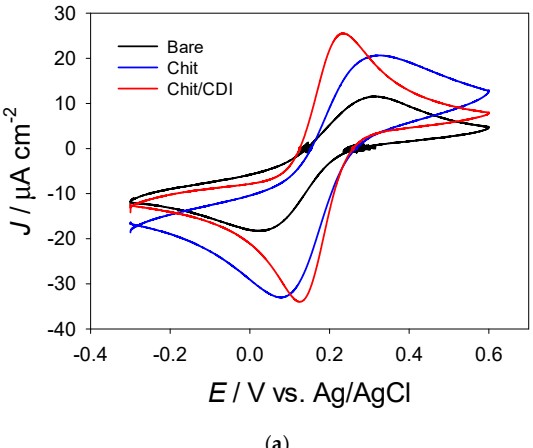

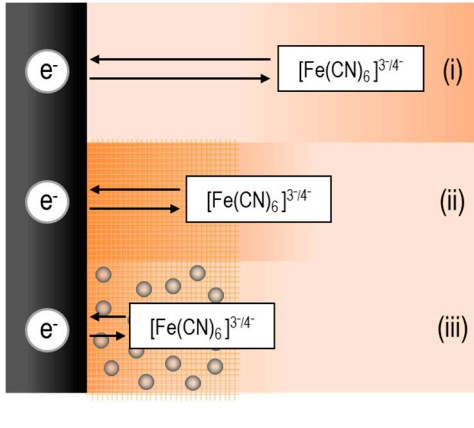

(**a**)  (**b**)

**Figure 4.** Evidence of electrostatic interaction enhancement between modified electrode and redox active species. (**a**) Cyclic voltammograms of bare and modified glassy carbon electrode (GCEs) in 0.1 M sodium phosphate buffer (pH 7.0) containing 1 mM of $[Fe(CN)_6]^{3-}$. Scan rate, 20 mV s$^{-1}$ (versus Ag/AgCl). Bare indicates bare GCE. Chit indicates chitosan-modified GCE. Chit/CDI indicates chitosan/carbodiimide compound-modified GFE. (**b**) Schematic illustration of the redox reactions of diffusing and confined $[Fe(CN)_6]^{3-}$ ions in a chitosan layer. (i) Bare electrode, (ii) chitosan-modified (GFE/Chit), and (iii) chitosan/carbodiimide compound-modified electrode.

### 3.3. MES Operation and Biofilm Morphology of Biocathodes

The bare and modified graphite cathodes were integrated with the microbial electrosynthesis (MES) reactor for $CO_2$ reduction to bioproducts (Figure 2). In this study, the $CO_2$ conversion efficiency was estimated by measurement of *R. sphaeroides* growth because the $CO_2$ fixed in microorganisms triggers a direct increase in their biomass via

growth [23,24]. The production of succinic acid, which is a robust organic product of $CO_2$ conversion in *R. sphaeroides*, was also considered the target product in this MES reaction [25,26]. The MES reactors were operated for 14 d under supplementation with 5% $CO_2$ at an applied potential of $-0.6$ V (vs. Ag/AgCl). Charge transfers from the working electrodes (cathodes) to the catholyte materials (*R. sphaeroides* and medium) reached 0.78, 0.93, and 0.94 mWh cm$^{-1}$ for bare, chitosan-modified, and chitosan/carbodiimide compound-modified GFE, respectively.

Figure 5 shows scanning electron microscope (SEM) images of bare (a,d) and modified GFEs (b,c,e,f) before (a–c) and after MES reactions (d–f). Sheath-like thin layers were observed at chitosan-coated GFE cathodes (b,c). *R. sphaeroides* cells attached to the cathode surfaces were remarkably increased following the order: bare < chitosan-modified < chitosan/carbodiimide-modified GFE. The results support the hypothesis that the modified cathode, especially chitosan/carbodiimide-modified GFE, facilitated the migration of *R. sphaeroides* (Figure 3), due to their positive and covalent functionality [13].

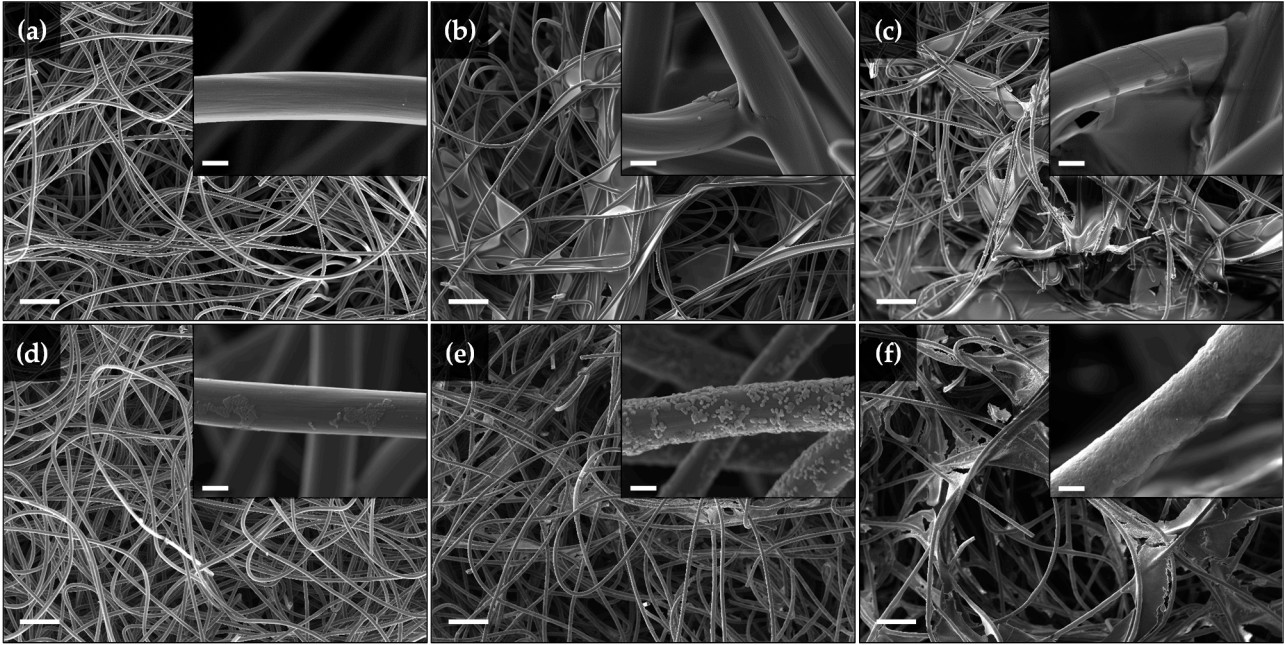

**Figure 5.** Scanning electron microscope images of graphite felt electrodes (GFEs) as cathodes before (**a–c**) and after (**d–f**) microbial electrosynthesis reaction by *R. sphaeroides* for 14 d. (**a,d**) bare GFE, (**b,e**) chitosan-modified GFE, (**c,f**) chitosan/carbodiimide compound-modified GFE. Scale bar, 100 μm (inset, 5 μm).

### 3.4. Enhanced $CO_2$ Conversion by Modified Cathodes

After 14 d operation of the MES, the final biomass of *R. sphaeroides* and succinic acid production were analyzed (Figure 6). Although the planktonic biomass (blue squares) decreased, the attached biomass (red circles) in the MES reactors equipped with two modified GFE cathodes increased substantially. The calculations of total mole concentrations, which sum up the planktonic and attached biomass, were as follows: 1.08, 2.25, and 2.64 mol for the bare, chitosan-modified, and chitosan/carbodiimide-modified GFE cathode, respectively. In addition, the largest amount of succinic acid production (bar graph) was observed at the MES reactor equipped with the chitosan/carbodiimide-modified GFE cathode (33.4 μmol). The faradaic efficiency of the chitosan/carbodiimide-modified GFE cathode-equipped MES reaction was significantly higher (98.0%) than that of the bare (34.9%) and chitosan-modified GFE cathode (77.8%), respectively. Zhang et al. reported that a modified cathode with chit/CDI in MES by using *S. ovata* as microbial catalyst increased acetate production 7.6-fold higher than the sample without cathode modification with Faradaic efficiency $86 \pm 12\%$ [12]. On the other hand, our study is shown to

have better performance attributed to chit/CDI modified cathode, triggering succinic acid production as much as 33.4 μmol with faradaic efficiency by 98%.

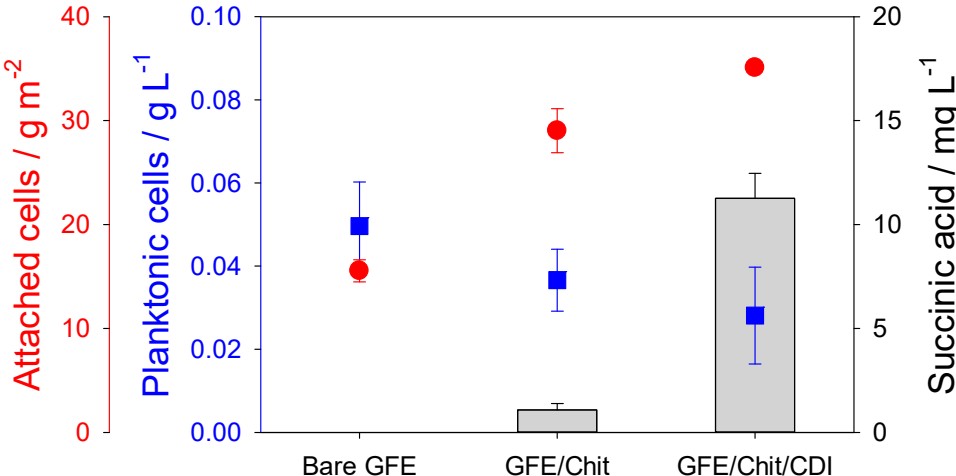

**Figure 6.** Final biomass (attached and planktonic cells) and succinic acid production from microbial electrosynthesis reaction of *R. sphaeroides* for 14 d under −0.6 V (vs. Ag/AgCl) and 5% $CO_2$. The weight of attached cells (g m$^{-2}$) was normalized to the area of the graphite felt electrode. Bar chart, succinic acid (mgL$^{-1}$). Square symbol (blue), planktonic cells (gL$^{-1}$). Circle symbol (red), attached cells (g m$^{-2}$).

Some electroactive bacteria can obtain electrons from an electrode directly through physical contact between the cathode and the microbial electron transport system [27]. This physical contact was developed in the form of a biofilm on the cathode [1]. For example, *Geobacter sulfurreducens*, a well-known electroactive microorganism, was genetically engineered to enable autotrophic growth by introducing genes for an ATP-dependent citrate lyase (it is named *G. sulfurreducens* strain ACL) [27]. The designed strain grew thicker confluent biofilms (ca. 35 μm) on graphite cathodes, resulting in higher current consumption (≥10-fold) than that of the wild type. Recently, Li et al. reported that *R. sphaeroides* produces $H_2$ with its biomass growth through direct electron transfer from cathode to bacteria in the MES reactor [7]. Taken together, these may reflect that cathode modification using chitosan/carbodiimide composite may facilitate electron utilization by improving direct contact between an electrode and *R. sphaeroides*.

## 4. Conclusions

The results presented in this paper illustrate that amide-coupling between a cathode and the $CO_2$-fixing microorganism, *R. sphaeroides*, improves MES reaction. A graphite felt cathode was modified with a combination of chitosan and a carbodiimide compound. Negatively charged *R. sphaeroides* cells (ζ-potential, −31.88 ± 3.31 mV) facilitated the formation of biofilm on positively charged modified cathodes. In particular, a robust biofilm of *R. sphaeroides* was developed on the chitosan/carbodiimide compound-modified cathode. Therefore, a chitosan/carbodiimide compound-modified (GFE/Chit/CDI) cathode was adopted for the optimal condition of cathode modification in this study. This modification enhanced the biomass and succinic acid production from $CO_2$ conversion in a MES reactor (applied potential at −0.6 V vs. Ag/AgCl). The calculated faradaic efficiency of the MES reactor equipped with a chitosan/carbodiimide compound-modified cathode was 98.0% (coulomb to *R. sphaeroides* biomass and succinic acid). This is a promising starting point for the development of future MES-driven $CO_2$ biorefinery process using *R. sphaeroides* to produce various spectra of metabolites such as biofuels and platform chemicals.

**Author Contributions:** Conceptualization, S.Y.L. and H.N.F.; methodology, J.L.; software, J.S.; validation, S.L., M.M. and Y.-K.O.; formal analysis, Y.R.L.; investigation, H.N.F. and M.P.; resources, J.L.; data curation, S.Y.L.; writing—original draft preparation, H.N.F.; writing—review and editing, S.Y.L.; visualization, J.L.; supervision, S.Y.L.; project administration, J.-S.L.; funding acquisition, J.-S.L. All authors have read and agreed to the published version of the manuscript.

**Funding:** This work was conducted under the framework of the Research and Development Program of the Korea Institute of Energy Research (KIER-C1-2432). This research also supported by the R&D Innovation cluster of the Republic of Korea (2021-DD-RD-0033-01-202).

**Institutional Review Board Statement:** Not applicable.

**Informed Consent Statement:** Not applicable.

**Data Availability Statement:** Not applicable.

**Conflicts of Interest:** The authors declare no conflict of interest.

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
