# Peer review of "Surface Modification of a Graphite Felt Cathode with Amide-Coupling Enhances the Electron Uptake of Rhodobacter sphaeroides"

_applsci, doi:10.3390/app11167585_

Round 1
Reviewer 1 Report
This MN applied novel graphite felt cathode to enhance the energy production from R. sphaeroides. It is an interesting topic on MEC. For inprovement this MN, some comments are shown as follows:
1) L66-72 The purpose of this study should be addressed clearly. What is the index for analyzing the performance of this study?
2) L112 Please show the wavelength of the LED lamp.
3) L130-131 Please address clearly how to take the biofilm on the cathode.
4) L210 The line types in Figure 3 can not be identified easier.
5) L227 Figure 4 should be moved to the "Materials and methods"section.
6) L247 What is the unit for "1.08, 2.25 and 2.64"? Are these data shown in Fig 6?
7) L254 Please show the legend in Fig 6.
8) Please compare the performance in this study to the literature.
9) L271 In the description in the "Conclusion", could you suggest the optimal condition according to the results of this study. The cumulative biomass in the cathode is good or bad in this study.
Author Response
Response to Reviewer 1’s Comments
Point 1: L66-72 The purpose of this study should be addressed clearly. What is the index for analyzing the performance of this study?
Response 1: In this study, we perform the cathode modification for MES by R. sphaeroides to increase bioproducts synthesis and Faradaic efficiency. In order to promote direct electron uptake, the graphite felt material was coated with a chitosan layer. The chitosan layer presents a unique functionalities (co-existing polycationic and neucleophilic properties) that are suitable for constructing a rigid biofilm onto the electrode surfaces (Lee et al., Appl. Sc., 2019). Additionally, the chitosan layer was modified with a carbodiimide compound to enable more substantial interaction between the R. sphaeroides cells and cathode surface (See Figure 1 in the manuscript).
Point 2: L112 Please show the wavelength of the LED lamp
Response 2: All MES reactors were placed under white LED lamp is in the range of 450-475 nm (www.photonic.com) at 28 ± 3 °C, which provided 50 μmol photons m-2 s-1. The reactors were operated in triplicate.
Point 3: L130-131 Please address clearly how to take the biofilm on the cathode
Response 3: The quantity of cells attached cells to the graphite felt cathode was determined by a gravimetric method for each biocathode. The weight of dried cathode before and after MES operation were weighed. A mass change of dried cathode after MES operation was calculated as biofilm weight (please see Lines 136 to 138 in the manuscript).
Point 4: L210 The line types in Figure 3 can not be identified easier.
Response 4: We have revised Figure caption to be identified easier as below (Lines 217 to 222 in the manuscript):
Figure 4 (in MS). Evidence of electrostatic interaction enhancement between modified electrode and redox active species. (a) Cyclic voltammograms of bare and modified glassy carbon electrode (GCEs) in 0.1 M sodium phosphate buffer (pH 7.0) containing 1 mM of [Fe(CN)6]3-. Scan rate, 20 mV s-1 (versus Ag/AgCl). Bare indicates bare GCE. Chit indicates chitosan-modified GCE. Chit/CDI indicates chitosan/carbodiimide compound-modified GFE. (b) Schematic illustration of the redox reactions of diffusing and confined [Fe(CN)6]3- ions in a chitosan layer. (i) Bare electrode, (ii) chitosan-modified (GFE/Chit), and (iii) chitosan/carbodiimide compound-modified electrode.
Point 5: L227 Figure 4 should be moved to the "Materials and methods"section
Response 5: Yes, Thank you for the response. It has been moved accordingly. Please check line 114 in the manuscript.
Point 6: L247 What is the unit for "1.08, 2.25 and 2.64"? Are these data shown in Fig 6?
Response 6: The unit for "1.08, 2.25 and 2.64" is mol which has been mentioned also in the text. This number is the sum of attached cells (gm-2) plus planktonic cells (gL-1). Please check the lines from 250 to 252 in the manuscript.
Point 7: L254 Please show the legend in Fig 6.
Response 7: We have described the legend carefully in Fig 6 as below (Lines 259 to 263):
Figure 6. Final biomass (attached and planktonic cells) and succinic acid production from microbial electrosynthesis reaction of R. sphaeroides for 14 d under -0.6 V (vs. Ag/AgCl) and 5% CO2. The weight of attached cells (g m-2) was normalized to the area of the graphite felt electrode. Bar chart, succinic acid (mgL-1). Square symbol (blue), planktonic cells (gL-1). Circle symbol (red), attached cells (gm-2).
Point 8: Please compare the performance in this study to the literature.
Response 8: Zhang et al (Energy Environ. Sci., 2013, 6, 217) reported that a modified cathode with chit/CDI in MES by using S. ovata as microbial catalyst increased 7.6 fold higher acetate production than sample without cathode modification with faradaic efficiency 86±12%. On the other hand, our study is shown to have better performance attributed to chit/CDI modified cathode triggering succinic acid production as much as 33.4 µmol with Faradaic efficiency by 98%.
We have add the discussion in the manuscript (please see lines from 257 to 262).
Point 9: L271 In the description in the "Conclusion", could you suggest the optimal condition according to the results of this study. The cumulative biomass in the cathode is good or bad in this study.
Response 9: The optimal condition of cathode modification in this study is chitosan/carbodiimide compound-modified (GFE/Chit/CDI) cathode. The results presented in this paper illustrate that amide-coupling between a cathode and the CO2-fixing microorganism, R. sphaeroides, improves MES reaction. For MES reaction more cumulative biomass on the cathode surface is good because the bacteria cells can uptake electron directly from the cathode. The final impacts of this phenomenon are increasing bioproducts synthesis and faradaic efficiency of MES.
We have revised “Conclusion” part according to the reviewer’s suggestion (please see lines from 288 to 291).

Reviewer 2 Report
This manuscript reports that Surface modification of graphite felt cathode with chitosan/carbodiimide enhances the electron uptake of Rhodobacter sphaeroides. In general, this manuscript is well written and organized. The results are interesting can could be useful for the future development of MES. Here are some minor comments:
1. What are the molecular weight and the degree of deacetylation of chitosan used in this study?
2. How much chitosan was incorporated into the graphite felt?
3. The author should do some analysis to prove the formation of amide coupling between the modified felt and the bacteria. Could it be that most of the bacteria are just physically attached to the modified felt?
4. What is the long-term efficiency of the modified cathode? The author might need to monitor the MES efficiency over a certain period.
5. In fig 6 and the corresponding paragraph. Why was the production of succinic acid using GFE/Chit/CDI around 11X to use GFE/Chit even if the attached biomass was only 5 cells / g mˆ2 different.
Author Response
Response to Reviewer 2’s Comments
Point 1: What are the molecular weight and the degree of deacetylation of chitosan used in this study?
Response 1: In this study, we used low molecular weight chitosan (50–190 kDa, 75-85% DDA) which was purchased from Sigma-Aldrich (St. Louis, MO) (please see Line 124 in the manuscript). Yuan et al. (Materials, 2011) reported the degree of deacetylation (DDA) and molecular weight (MW) of chitosans are important to their physical and biological properties such as crystallinity, hydrophilicity, degradation and cell response. Aside from that, Freier et al. (Biomaterials, 2005) found that prolonged degradation times and enhanced cell adhesion can be achieved using chitosan with a DDA close to 0% or 100% while chitosans with intermediate DDAs exhibit rapid degradation rates, but at the cost of limited cell adhesion.
Point 2: How much chitosan was incorporated into the graphite felt?
Response 2: We immersed the graphite felt electrode in the 2% chitosan overnight so that the chitosan can be attached to the electrode surface area more evenly (40 cm2), but we could not calculate the amount (ml) of incorporated chitosan into graphite felt. Because applied chitosan formed really thin layer onto the graphite felt fibres as shown in Figure 5 in the manuscript.
Point 3: The author should do some analysis to prove the formation of amide coupling between the modified felt and the bacteria. Could it be that most of the bacteria are just physically attached to the modified felt?
Response 3: Higher faradaic efficiency and organic acid biosynthesis from GFE/Chit/CDI modified cathode than GFE/Chit modified cathode is proof that there is also biochemical interaction, besides physically attached phenomenon, between bacteria cells and modified cathode surface which enhance the transfer of electron from cathode to bacteria cells.
Although the direct analysis is still out of scope in this work, thereby it is necessary for the next researcher to investigate direct implication of the bacteria attachment.
Point 4: What is the long-term efficiency of the modified cathode? The author might need to monitor the MES efficiency over a certain period
Response 4: In the author's perspective the long-term efficiency in our MES is represented by Faradaic efficiency which is collected data from 14 days MES experiment. So, in this case, the Faradaic efficiency for chitosan/carbodiimide-modified electrode for 14 days is 98%.
Point 5: In fig 6 and the corresponding paragraph. Why was the production of succinic acid using GFE/Chit/CDI around 11X to use GFE/Chit even if the attached biomass was only 5 cells / g mˆ2 different.
Response 5: Chit/CDI crosslinking serves biochemical interaction between the cathode surface and the bacteria cell, it also provides stable porous (reference) chitosan that enhanced the adhesion physically. Therefore, the electron transfer was more efficient than that one from the sample without Chit/CDI crosslinking. The final impacts of this phenomenon are increasing bioproducts synthesis and faradaic efficiency of MES.

Reviewer 3 Report
This is an excellent and well presented paper describing the performance of cathodes modified by R. sphaeroides for use in microbial electrosynthesis for converting CO2 and renewable electrical energy into chemicals and fuels for the possible storage of solar and wind energies.
In lines 240, 243 and 256 of the manuscript, it should be 14 days instead of 14 d .
Author Response
Response to Reviewer 3’s Comments
Comments and Suggestions for Authors
This is an excellent and well presented paper describing the performance of cathodes modified by R. sphaeroides for use in microbial electrosynthesis for converting CO2 and renewable electrical energy into chemicals and fuels for the possible storage of solar and wind energies.
Response: Your consideration is much appreciated.